# Healthcare utilization for atopic dermatitis: An analysis of the 2010–2018 health insurance review and assessment service national patient sample data

**Sowon Kim**[1‡], **Ye-Seul Lee**[2‡], **Jiyoon Yeo**[2], **Donghyo Lee**[3], **Dong Kun Ko**[4], **In-Hyuk Ha**[2]*

**1** Jaseng Hospital of Korean Medicine, Gangnam-daero, Gangnam-gu, Seoul, Republic of Korea, **2** Jaseng Spine and Joint Research Institute, Jaseng Medical Foundation, Gangnam-daero, Gangnam-gu, Seoul, Republic of Korea, **3** Department of Ophthalmology, Otolaryngology, and Dermatology, College of Korean Medicine, Woo-Suk University, Jeonju, Korea, **4** Jayeonsaeng Korean Medicine Clinic, Yongin, Korea

‡ SK and YSL are co-first authors on this work.
* hanihata@gmail.com

**Data Availability Statement:** The datasets generated during and/or analyzed during the current study are available in the HIRA-NPS repository upon request [http://opendata.hira.or.kr]

## Abstract

This cross-sectional, retrospective, observational study aimed to analyze the distribution and healthcare usage patterns of patients with atopic dermatitis using the 2010–2018 Health Insurance Review and Assessment Service data. Patients diagnosed with atopic dermatitis in Korea between January 2010 and December 2018 and registered in the Health Insurance Review and Assessment national database were identified, and 270,008 patients who used healthcare services at least once during this period were evaluated to ascertain the healthcare usage patterns and treatment methods for atopic dermatitis. The number of patients with atopic dermatitis plateaued during the study period, while the number of claims and total expenses increased by a small margin. Atopic dermatitis prevalence was the highest among patients aged <5 years (31.4%), followed by those aged 5–14 years (23.53%) and 15–24 years (15.33%). However, the prevalence in these age groups showed a decreasing trend over time. The most used Western medicine treatments were injections and oral medications involving topical corticosteroids, antihistamine agents, and oral steroids, while it was acupuncture therapy in Korean medicine. The frequency of the most frequently prescribed medication, topical corticosteroid, showed a decreasing trend over time. The findings in this study will inform healthcare policy makers and clinicians across different countries on the usage trends of Western medicine and Korean medicine treatment.

## Introduction

Atopic dermatitis (AD) is a chronic and relapsing inflammatory condition that presents with erythematous, scaly, itchy rashes. It is most prevalent in children, but all age groups can be affected [1, 2]. Recent evidence indicates that it causes changes in the skin barrier and impairs immune function, increasing vulnerability to this disease throughout lifetime [3, 4]. AD exacerbation can lead to atopic march, defined as a natural history of atopic symptoms that

and upon payment of a data request fee (300 000 KRW per dataset).

**Funding:** This research was funded by Jaseng Medical Foundation, Republic of Korea, grant number JS-RP-2021-21. The funders had no role in study design, data collection and analysis, decision to publish, or preparation of the manuscript.

**Competing interests:** The authors declare no conflict of interest.

include AD and other allergic diseases (e.g., allergic asthma and allergic rhinitis) that may appear in early childhood [5, 6]. AD is a common disease, with a global prevalence of 10%–20% [7]. The incidence of atopic diseases has drastically increased over the past decades [8], and AD is one of the most common inflammatory skin diseases in developed countries. In total, 1%–3% of European adults have AD [9, 10]. In addition, the incidence is increasing in Europe and the US [11]. A 2015 study on pediatric AD patients (age < 18 years) reported a prevalence of 15.5%, with AD being more prevalent in girls. In addition, the prevalence was higher among children aged 3–9 and 10–14 years than among those aged 0–2 years, with patients having a mean age of 7.72 years. Meanwhile, the incidence was the lowest among children aged 15–17 years. Further, AD was more prevalent in urban areas and less prevalent in the lowest income group [12].

A high proportion of AD patients have severe disease, and AD can also occur in adults despite having a predilection for children. Importantly, AD can cause various disabilities, thus incurring direct and indirect expenses [2]. Consequently, AD imposes a significant economic burden at the personal and societal levels [13–16]. Direct expenses for AD in the US were approximately $364 million and $3.8 billion in 1993 and 2002, respectively, indicating increasing costs [17]. Similar to other chronic pediatric diseases, AD can have adverse effects on the social, personal, and emotional quality of life of both patients and families [18]. From a personal health perspective, it can also have a negative impact by causing chronic skin diseases such as urticaria and alopecia; sleep disturbance due to itching; as well as complications of comorbid asthma and food allergies [18]. From the social and emotional perspectives, it can cause behavioral and maladjustment problems as well as abnormal psychological development [18]. The impacts on family members include unstable relationships with spouses and other family members [18].

Oral medications including antihistamines, topical corticosteroids (TCS), oral steroids, and topical immunomodulators are the most frequently used treatments for AD [19]. A long-term therapy plan that avoids corticosteroid adverse effects while enhancing their efficacy has always been the subject of controversy. Systemic immunosuppressants e.g., cyclosporine, methotrexate, and mycophenolate mofetil are recommended for severe AD, while caution is warranted to avoid side effects when long-term treatments are administered [19, 20]. One survey conducted to Korean dermatologists stated that systemic immunosuppressants are not actively used in clinical settings [21].

In this context, the use of CAM in dermatology is gaining attention [22]. In a study published in 2016 on the management of AD in Korea, 55.9% of subjects had used complementary and alternative medicine (CAM), with Korean medicine (KM) being the most common (72.4%). The primary reason for using CAM was the chronic progression of AD, and concerns about adverse events (AEs) associated with TCS [23]. A US study showed that CAM is often used because of concerns about AEs associated with conventional treatment [24], and patients who use CAM are mainly female and Caucasian. A study in Denmark also showed that CAM is often used in children and adults with long-term disease morbidity and serious disease burden [25]. While the effectiveness of CAM on AD has not been confirmed [26, 27], a few clinical trials and reviews have reported possible efficacy of acupuncture for symptom improvement and reduced recurrence across different populations [28–33], and one study reported a trend towards reduced TCS use in acupuncture and osteopathic treatment groups [34]. Nonetheless, it is crucial for both physicians and patients to note the uncertainties involved in CAM therapies due to lack of evidence on its safety and efficacy in AD management. The potential risk of inappropriate use of CAM which may induce allergic reactions, additional burden of cost, and possible negligence of conventional medication by patients which may accelerate disease progress needs to be taken into account when analyzing utilization of CAM.

The single insurer system in Korea by National Health Insurance (NHI), which supports 98% of all population and also provides data for 2% of the population who receive Medical Aid, provides healthcare coverage in both Western and Korean Medicine due to its unique dual healthcare system. Therefore, examining the NHI database provides knowledge on healthcare spendings from societal perspective in both Western and Korean Medicine. Data from the Korean NHI database shows decreasing incidence but increasing treatment cost of AD. In total, 1,035,680 patients with AD (KCD-10 code L208/209) were treated in 2010, and the total treatment cost was approximately $53,963,917 (KRW1,156/USD, 2010 yearly average exchange rate). In 2019; 949,351 patients were treated, and the total cost was approximately $74,776,603 (KRW1,166.1/USD, average exchange rate for 2019), implying a decrease in the number of patients but an increase in total cost [35]. A 2015 study showed that the direct expenses associated with AD and expenses for other AD-related products per patient were 541,280 KRW and 120,313 KRW, respectively, during the 3-month study period. The esti-mated annual direct expense (including expenses associated with other AD-related products) per patient was 2,646,372 won, while the estimated annual indirect expense was 1,507,068 won. The annual disease cost (direct and indirect expenses) of AD was estimated to be 4,153,440 won. The total social cost of AD at national level is estimated to be 5.8 trillion won per year [36].

The plateauing number of patients with AD and increasing expenses highlight the need for guidelines that reflect the current prevalence and treatment patterns of AD at the national level. However, insufficient data on AD in Korea necessitates further studies on the type of treatment universally administered and the associated costs. Moreover, with a dual healthcare system in Korea that allows patients to choose between Western medicine (WM) and KM for the same disease, there is a need for basic research that reflects the characteristics of different types of medicine to understand the healthcare situation in Korea. Accordingly, the present study aimed to perform a comparative analysis of AD status, treatment, and route of access according to KM and WM among AD patients in Korea. Towards this goal, we analyzed data from the Health Insurance Review and Assessment (HIRA) Service claims from 2010 to 2018.

## Materials and methods

### Data source

The study used HIRA National Patient Sample (HIRA-NPS) data covering a 9-year period from 2010 to 2018. Health insurance claim data are generated when a healthcare facility files a claim for reimbursement to the HIRA for healthcare services provided to a patient. The HIRA provides random stratified data in 1-year increments for data accessibility and research conve-nience. The data used in this study was secondary data, statistically sampled after removal of personal and corporate information from raw data. It consisted of services claimed during a 1-year period from the commencement of care for the applicable year. Details of treatments and prescriptions for all patients who used the healthcare service during the 1-year period are included, and patients are sampled by stratified, systematic sampling according to sex and age group (10-year increments). Each year, 3% of all patients are randomly sampled (Fig 1) [37].

### Study design and population

This study evaluated patients with a main diagnosis of AD (KCD-10 code L208/209) during the applicable period, including patients of all ages who were treated at least once for AD. Among the 636,655 records initially identified, we excluded those with a form code indicating dentistry, health center, or psychiatry (n = 2,994); treatment at a nursing hospital, psychiatric hospital, dental hospital, postpartum care facility, or health center (n = 643); and total expenses

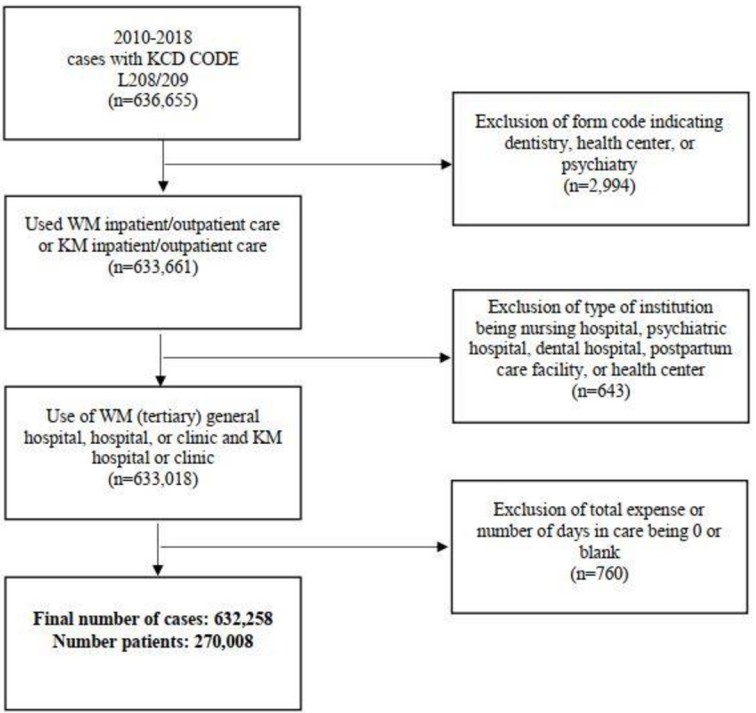

**Fig 1. Flowchart of record selection process.**

or number of days in care of 0 or missing (n = 760). Consequently, a total of 632,258 treatment records of 270,008 patients were analyzed (Fig 1).

## Statistical analysis

The number of patients, total cases, total expenses, per-patient expenses, per-case expenses, total care, average days in care per patient, total visits, and average visits per patient were analyzed according to WM, KM, and overall data by year. The treatment received during outpatient visits to WM and KM institutions, frequency of such treatment, average per-visit expenses, and average total cost per year were described. In addition, graphs were generated to examine annual changes. The patients were classified by age (eight categories in 10-year increments from <5 to ≥65 years), sex, and insurance type (national health insurance, Medicaid, and others), and the frequency of each was analyzed.

In accordance with the HIRA guidelines from the Ministry of Health and Welfare, total expenses were divided into eight categories of consultation, injection, hospitalization, testing, medication and prescription, treatment and surgery, diagnostic radiology or radiation therapy, and others. "Injection fee" included the medication being injected and the method of injection. Acupuncture and pharmacopuncture, which are invasive treatments with needles inserted into acupoints, were included as injections in the analysis of KM syndrome differentiation. Total expense was the sum of out-of-pocket expenses and the contribution from the National Health Insurance Service for charges incurred by patients with health insurance. AD-related service codes were divided by WM and KM and classified, based on items that accounted for at least 0.1% among all WM and KM cases (WM: top 145 items; KM: top 75 items). For each item, total cases, total expense, number of patients, average annual cases per patient, and average annual expense per patient were analyzed. The frequency of claims and expenditure for

each item was also analyzed, and the items were arranged from the highest to lowest frequency. The average annual log change for each item, and the differences between WM and KM, were investigated [38, 39].

Medications prescribed during inpatient and outpatient visits were categorized according to the Anatomical Therapeutic Chemical classification code by referring to the Ministry of Health and Welfare classification standard. The medication codes were reported by frequency according to the medication classification (basic, adjuvant, or optional treatment) [29], average expense per visit, and average total annual cost per patient. Packaged herbal medicine was excluded from the analysis because it is not included in the list of accredited medications for KM. A graph was generated to examine annual changes.

All expenses were converted to USD from KRW according to the exchange rate for the applicable year and adjusted based on the health sector consumer price index for 2018 (S1 Table). All statistical analyses were performed using SAS 9.4 (2002–2012 by SAS Institute Inc., Cary, NC, USA).

### Ethics approval

Ethical review and approval were waived for this study by the Institutional Review Board of Jaseng Hospital of Korean Medicine in Seoul, Korea (JASENG 2021-07-008), because all personal data were deidentified by the HIRA-NPS. Patient consent was waived because we used publicly available data and the information of the subjects was deidentified.

## Results

### Trends in incidence and expense

As shown in Table 1, a total of 32,758 patients visited a medical institution for AD in 2010; among them, 31,498 and 1,260 patients received WM and KM, respectively. In 2018, 28,739 patients were treated; among them, 27,814 and 925 patients received WM and KM, respectively, showing a decreasing trend with a mild slope. The number of patients who visited a medical institution for AD throughout the study period decreased. With respect to treatment type, the number of patients treated with WM was significantly higher, by approximately 24–30 times, than the number of patients treated with KM. There were 70,073 claims in 2010, and there were significantly more claims for WM (62,412 vs 7,661 KM). Meanwhile, there were 70,732 claims in 2018, with no significant difference from that in 2018. Among these claims, 63,294 and 7,438 were for WM and KM, respectively. In contrast, the total expense increased from $866,232.15 ($769,556.20 in WM and $96,675.95 in KM) in 2010 to $1,362,125.02 ($1,212,532.21 in WM and $149,592.81 in KM) in 2018. The average per-patient expense also increased for both WM and KM, but with an especially sharp increase in KM-related expenses (Fig 2). Moreover, per-patient expense was 3–4 times higher for KM than for WM. In contrast to the decreasing trend in number of patients, the number of claims did not show a decreasing trend. Furthermore, treatment expenditure, number of visits, and per-patient expenses showed an increasing trend.

### Patient characteristics

With respect to incidence, the overall number of patients decreased, with a mild slope (12% decrease). The steepest decrease was found in the <5 years group, and a decreasing trend was also found in the 5–14 years group, indicating that incidence decreased most among pediatric patients. Patients aged <14 years accounted for 62% of all patients in 2010, but this proportion decreased sharply to 46% in 2018. The percentage of patients aged ≥24 years showed a

**Table 1. Medical service usage for AD in Korea.**

| Year | Type of visit | Number of patients | Total claims | Total expense | Per-patient expense | Per-claim expense | Total care | Ave. days in care per patient | Total visits | Ave. days visits per patient |
|---|---|---|---|---|---|---|---|---|---|---|
| 2010 | Total | 32,758 | 70,073 | 866,232.15 | 26.44 | 12.36 | 77,036 | 2.35 | 72,912 | 2.23 |
| | WM | 31,498 | 62,412 | 769,556.20 | 24.43 | 12.33 | 68,608 | 2.18 | 65,087 | 2.07 |
| | KM | 1,260 | 7,661 | 96,675.95 | 76.73 | 12.62 | 8,428 | 6.69 | 7,825 | 6.21 |
| 2011 | Total | 32,128 | 69,444 | 946,068.22 | 29.45 | 13.62 | 77,567 | 2.41 | 71,996 | 2.24 |
| | WM | 30,850 | 61,222 | 830,493.48 | 26.92 | 13.57 | 68,782 | 2.23 | 63,592 | 2.06 |
| | KM | 1,278 | 8,222 | 115,574.74 | 90.43 | 14.06 | 8,785 | 6.87 | 8,404 | 6.58 |
| 2012 | Total | 30,715 | 67,460 | 912,741.15 | 29.72 | 13.53 | 75,012 | 2.44 | 67,749 | 2.21 |
| | WM | 29,497 | 59,042 | 794,945.47 | 26.95 | 13.46 | 66,177 | 2.24 | 59,290 | 2.01 |
| | KM | 1,218 | 8,418 | 117,795.68 | 96.71 | 13.99 | 8,835 | 7.25 | 8,459 | 6.94 |
| 2013 | Total | 30,888 | 68,791 | 1,005,176.47 | 32.54 | 14.61 | 78,068 | 2.53 | 69,032 | 2.23 |
| | WM | 29,662 | 61,103 | 889,698.68 | 29.99 | 14.56 | 69,429 | 2.34 | 61,333 | 2.07 |
| | KM | 1,226 | 7,688 | 115,477.79 | 94.19 | 15.02 | 8,639 | 7.05 | 7,699 | 6.28 |
| 2014 | Total | 30,158 | 71,357 | 1,163,894.95 | 38.59 | 16.31 | 81,014 | 2.69 | 71,564 | 2.37 |
| | WM | 28,874 | 61,878 | 1,006,169.41 | 34.85 | 16.26 | 70,897 | 2.46 | 62,036 | 2.15 |
| | KM | 1,284 | 9,479 | 157,725.54 | 122.84 | 16.64 | 10,117 | 7.88 | 9,528 | 7.42 |
| 2015 | Total | 29,564 | 71,467 | 1,150,638.91 | 38.92 | 16.10 | 78,552 | 2.66 | 71,777 | 2.43 |
| | WM | 28,335 | 62,097 | 1,001,474.19 | 35.34 | 16.13 | 68,066 | 2.40 | 62,387 | 2.20 |
| | KM | 1,229 | 9,370 | 149,164.72 | 121.37 | 15.92 | 10,486 | 8.53 | 9,390 | 7.64 |
| 2016 | Total | 29,653 | 71,756 | 1,179,550.01 | 39.78 | 16.44 | 77,402 | 2.61 | 71,998 | 2.43 |
| | WM | 28,550 | 62,983 | 1,029,855.54 | 36.07 | 16.35 | 67,510 | 2.36 | 63,143 | 2.21 |
| | KM | 1,103 | 8,773 | 149,694.47 | 135.72 | 17.06 | 9,892 | 8.97 | 8,855 | 8.03 |
| 2017 | Total | 29,437 | 71,178 | 1,280,536.48 | 43.50 | 17.99 | 76,877 | 2.61 | 71,323 | 2.42 |
| | WM | 28,401 | 63,170 | 1,139,503.65 | 40.12 | 18.04 | 67,792 | 2.39 | 63,295 | 2.23 |
| | KM | 1,036 | 8,008 | 141,032.83 | 136.13 | 17.61 | 9,085 | 8.77 | 8,028 | 7.75 |
| 2018 | Total | 28,739 | 70,732 | 1,362,125.02 | 47.40 | 19.26 | 76,251 | 2.65 | 70,957 | 2.47 |
| | WM | 27,814 | 63,294 | 1,212,532.21 | 43.59 | 19.16 | 67,776 | 2.44 | 63,429 | 2.28 |
| | KM | 925 | 7,438 | 149,592.81 | 161.72 | 20.11 | 8,475 | 9.16 | 7,528 | 8.14 |

AD, atopic dermatitis; WM, Western medicine; KM, Korean medicine

All expenses are converted to USD from KRW according to the annual average exchange rate in the applicable year (see S1 Table).

gradually increasing trend (Fig 3). As shown in Table 2, there were more females (52.71%) than males (47.29%) who visited a medical institution for AD. Prevalence was the highest in the <5 years group (31.4%), followed by the 5–14 years (23.53%) and 15–24 years groups (15.33%), indicating a high prevalence among infants, children, and adolescents. Particularly, patients aged <14 years accounted for a high proportion of patients at 54.93%. The number of patients aged 15–24 years was almost double that of patients aged 25–34 years (41,384 vs 23,091).

WM usage was the highest in the <5 years age group (31.56%), followed by the 5–14 years (23.44%) and 15–24 years (15.09%) groups. Meanwhile, KM usage was highest in the 5–14 years group (25.68%), followed by the <5 years (22.9%) and 15–24 years groups (19.37%). Among WM users, 47.3% and 52.7% were male and female, respectively, while 44.16% and 55.84% of KM users were male and female, respectively, with no difference in sex distribution. However, there was a slightly higher proportion of males who used both WM and KM (51.41% vs 48.59%). For insurance type, 96.95% had NHI and 2.99% had Medicaid, and the patient distribution by insurance type was not significantly different between WM and KM. S2

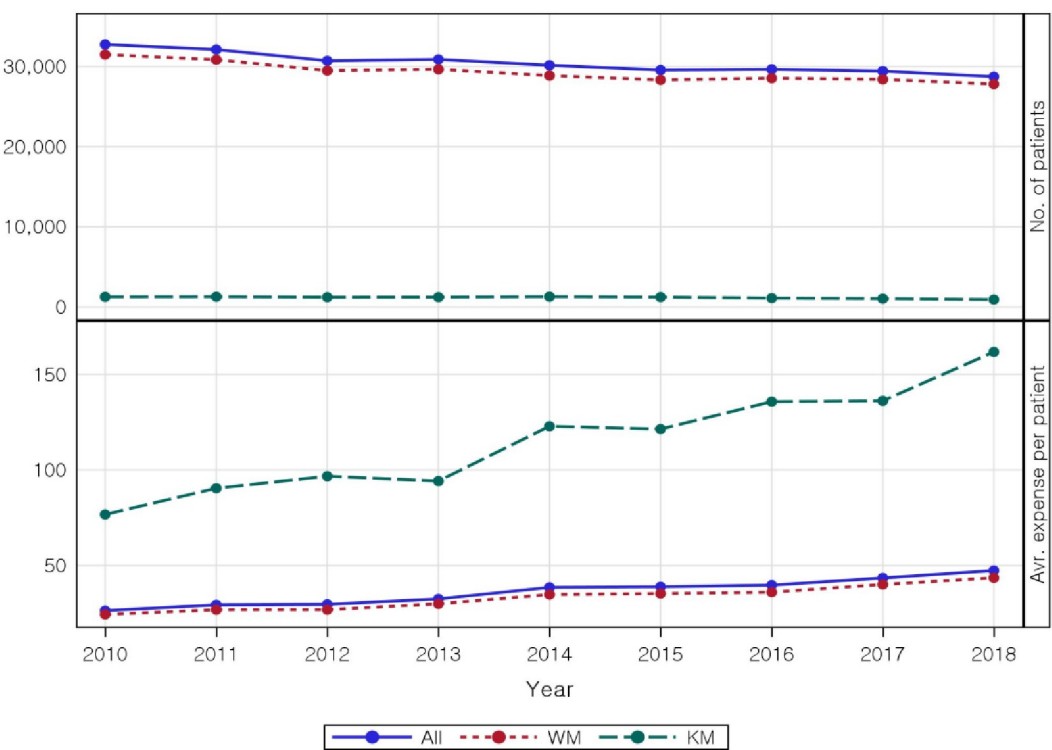

**Fig 2. Trends in the incidence of atopic dermatitis, average total visits per patient, and Korean medicine expenses.** WM: Western medicine; KM: Korean medicine.

Table describes the healthcare usage behavior of patients with AD. S3 Table describes the major sub-diagnoses of patients with AD indicating comorbidities of AD.

### Average rate of change in total expense and number of cases by item

The total expense and number of claims for AD treatment by year are shown in Table 3. For WM and KM combined, consultation fees accounted for the highest 9-year average total expense at $749,394.04. This increased by an average of 4.04% annually, and an average of 91,659.67 claims were filed each year. Testing fees accounted for the second highest average total expense at $188,941.17. This increased by an average of 9.97% annually, and an average of 20,535.00 claims were filed annually. The third highest expense was injection fees at $85,149.99. This increased by an average of 6.26% each year, and an average of 41,224.56 claims were filed annually. The fourth highest expense was treatment and surgery fees at $25,397.24. This increased by an average of 16.51% each year, and an average of 2,447.44 claims were filed annually. With respect to the 9-year average increase rate, treatment and surgery fees (16.51%) showed the highest increase, followed by other treatment fees (16.32%).

For WM only, the highest total expense was the consultation fees ($689,097.79), followed by testing ($188,941.17), treatment/surgery ($25,397.24), other treatment ($24,794.97), and injection fees ($15,621.92). For KM only, the highest total expense was for injection fees ($69,528.08), followed by consultation ($60,296.24) and hospitalization fees ($1,844.75). In KM, hospitalization fees showed the highest 9-year average increase, at 29.38% increase per year. Diagnostic radiology or radiation therapy fees and treatment/surgery fees were not included because they are not covered as KM benefits. Among the WM items, the highest

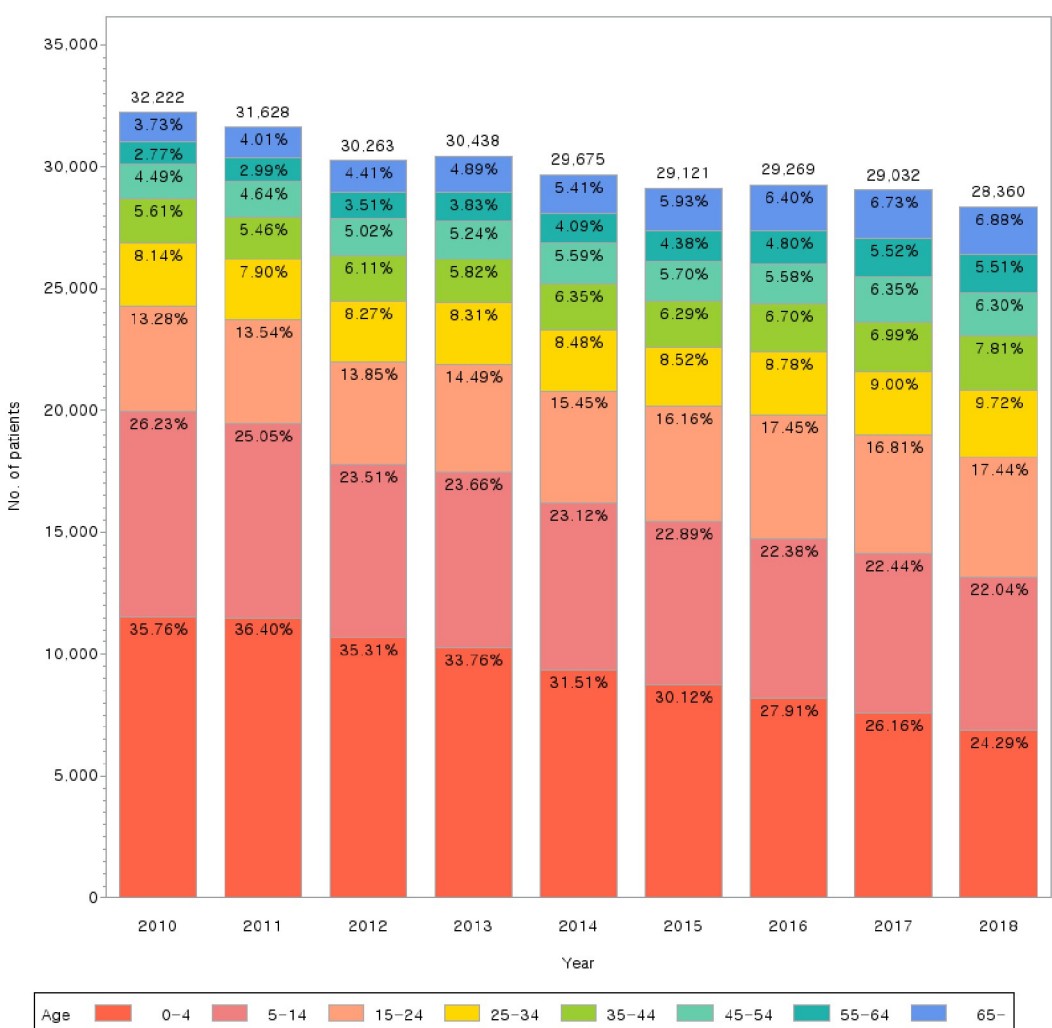

**Fig 3. Age trends in atopic dermatitis from 2010 to 2018.**

number of claims were for subcutaneous, intramuscular or intravenous injections
(n = 88,764). Meanwhile, per-case expense throughout the 9-year period was highest for allergy tests at 12,824 claims and per-case expense of $94.07. Allergy tests also showed the highest average annual total per-patient expense at $110.71. The second and third highest average per-patient expense was for other tests ($83.42) and complete blood count (CBC) test (Code B1~: $29.22), respectively, indicating that testing fees accounted for a large portion of the cost burden for patients. Among KM items, acupuncture showed the highest number of claims over the 9-year period at 71,185. The highest average expense per claim was for other treatments ($10.96), followed by that for acupuncture ($6.71), direct moxibustion ($6.17), and bloodletting cupping ($5.98). Moreover, the highest average annual total per-patient expense was for acupuncture ($53.32), followed by direct moxibustion ($50.84) and bloodletting cupping ($32.09), showing a similar tendency as average expense per claim, except for other treatments (Table 4).

Among the basic medication treatments for AD, TCS showed the highest number of claims at 340,187, accounting for approximately 91% of all basic medication treatments. Meanwhile,

**Table 2. Basic patient characteristics.**

| Category | | Patient | | | | | | | |
|---|---|---|---|---|---|---|---|---|---|
| | | Total (2010–2018) | | Only WM (2010–2018) | | Only KM (2010–2018) | | Both WM and KM (2010–2018) | |
| | | Total N | Percent | Total N | Percent | Total N | Percent | Total N | Percent |
| Age, years | <5 (years) | 84,775 | 31.40 | 81,886 | 31.56 | 1,495 | 22.90 | 1,394 | 34.57 |
| | 5–14 | 63,538 | 23.53 | 60,825 | 23.44 | 1,676 | 25.68 | 1,037 | 25.72 |
| | 15–24 | 41,384 | 15.33 | 39,155 | 15.09 | 1,264 | 19.37 | 965 | 23.93 |
| | 25–34 | 23,091 | 8.55 | 21,700 | 8.36 | 949 | 14.54 | 442 | 10.96 |
| | 35–44 | 17,078 | 6.32 | 16,433 | 6.33 | 511 | 7.83 | 134 | 3.32 |
| | 45–54 | 14,611 | 5.41 | 14,264 | 5.50 | 318 | 4.87 | 29 | 0.72 |
| | 55–64 | 11,130 | 4.12 | 10,940 | 4.22 | 170 | 2.60 | 20 | 0.50 |
| | ≥65 | 14,401 | 5.33 | 14,246 | 5.49 | 144 | 2.21 | 11 | 0.27 |
| Sex | Male | 127,674 | 47.29 | 122,719 | 47.30 | 2,882 | 44.16 | 2,073 | 51.41 |
| | Female | 142,334 | 52.71 | 136,730 | 52.70 | 3,645 | 55.84 | 1,959 | 48.59 |
| Insurance type | NHI | 261,771 | 96.95 | 251,370 | 96.89 | 6,433 | 98.56 | 3,968 | 98.41 |
| | Medicaid | 8,066 | 2.99 | 7,908 | 3.05 | 94 | 1.44 | 64 | 1.59 |
| | Others | 171 | 0.06 | 171 | 0.07 | - | - | - | - |

KM: Korean medicine; NHI, National Health Insurance; WM: Western medicine

although the usage rate for topical calcineurin inhibitors (TCI) was much lower than that for TCS, the average expense per case was $18.47, approximately 4.68 times higher than that for TCS. Among adjuvant medication treatments for AD, antihistamine agents showed the highest number of claims at 534,597 cases, accounting for approximately 86% of all claims for adjuvant medication treatments. This was followed by antibiotics for topical use & for systemic use, antifungals for topical use & antimycotics for systemic use, and gamma-linolenic acid. The highest average expense per case was for antifungals for topical use & antimycotics for systemic use, while the highest average per-patient expense was for gamma-linolenic acid. Among

**Table 3. Total and average change in expense and number of claims over the 9-year study period.**

| | All | | | | Western medicine | | | | Korean medicine | | | |
|---|---|---|---|---|---|---|---|---|---|---|---|---|
| | Total expense | | No. of claims | | Total expense | | No. of claims | | Total expense | | No. of claims | |
| | Ave. sum | Ave. CR* | Ave. n | Ave. CR* | Ave. sum | Ave. CR* | Ave. n | Ave. CR* | Ave. sum | Ave. CR* | Ave. n | Ave. CR* |
| Consultation fee | 749,394.04 | 4.04 | 91,659.67 | 3.07 | 689,097.79 | 4.03 | 82,580.22 | 3.20 | 60,296.24 | 4.06 | 9,079.44 | 1.70 |
| Injection fee | 85,149.99 | 6.26 | 41,224.56 | 0.75 | 15,621.92 | 4.80 | 20,728.44 | (0.18) | 69,528.08 | 6.61 | 20,496.11 | 1.75 |
| Hospitalization fee | 13,490.27 | 4.12 | 240.78 | 4.82 | 11,645.52 | 0.12 | 212.89 | 2.97 | 1,844.75 | 29.38 | 27.89 | 20.37 |
| Testing fee | 188,941.17 | 9.97 | 20,535.00 | 5.10 | 188,941.17 | 9.97 | 20,535.00 | 5.10 | 748.79 | (12.75) | 181.89 | (8.18) |
| Medication and prescription fee | 10,228.15 | 2.01 | 16,004.11 | (4.28) | 8,870.14 | 0.68 | 15,425.56 | (4.60) | 1,358.02 | 8.94 | 578.56 | 5.00 |
| Treatments and surgery | 25,397.24 | 16.51 | 2,447.44 | 9.42 | 25,397.24 | 16.51 | 2,447.44 | 9.42 | - | - | - | - |
| Diagnostic radiology or radiation therapy fee | 2,107.74 | 2.91 | 478.56 | (5.68) | 2,107.74 | 2.91 | 478.56 | (5.68) | - | - | - | - |
| Other treatment fee | 25,543.76 | 16.32 | 2,781.89 | 9.72 | 24,794.97 | 17.56 | 2,600.00 | 11.12 | - | - | - | - |

Ave. sum: 9-year average of total expense; Ave. n, 9-year average of number of cases; Ave. CR, 9-year average rate of change

All expenses are converted from KRW to USD according to the annual average exchange rate. The price level of health expenses is adjusted as of year 2018 (see S1 Table).

**Table 4. Frequency of outpatient interventions by type (WM and KM).**

| WM Outpatients | Total claims | Average cost per bill | Average cost per patient |
|---|---|---|---|
| Subcutaneous, intramuscular or intravenous injection | 88,764 | 1.43 | 3.21 |
| Photodynamic therapy | 14,695 | 13.87 | 65.11 |
| Allergy test | 12,824 | 94.07 | 110.71 |
| Wet-wrap therapy | 10,097 | 8.46 | 15.73 |
| CBC test (Code B1~) | 9,646 | 24.05 | 29.22 |
| Phototherapy | 4,954 | 2.38 | 4.58 |
| Nebulizer treatment | 2,278 | 2.35 | 3.94 |
| Urinalysis (Code B0~) | 1,919 | 2.46 | 3.12 |
| Other tests | 18,381 | 66.37 | 83.42 |
| Other treatment | 16,892 | 7.40 | 13.78 |
| KM Outpatients | Total claims | Average cost per bill | Average cost per patient |
| Acupuncture | 71,185 | 6.54 | 51.84 |
| Hot/cold meridian therapy- | 18,412 | 0.81 | 6.84 |
| Moxibustion (indirect moxibustion) | 12,996 | 2.37 | 16.28 |
| Cupping (dry cupping) | 9,089 | 3.72 | 25.58 |
| Cupping (bloodletting cupping) | 3,738 | 5.84 | 40.43 |
| Electroacupuncture stimulation | 2,127 | 4.01 | 28.75 |
| Herbal medicines | 1,495 | 4.20 | 4.64 |
| Moxibustion (direct moxibustion) | 1,113 | 6.09 | 71.30 |
| Other treatments | 5,972 | 10.68 | 11.68 |

All expenses are converted from KRW to USD according to the annual average exchange rate (see S1 Table).

KM: Korean medicine

optional prescription medications, oral steroids showed the highest number of claims at 163,811, accounting for approximately 47% of all optional prescriptions. Despite the average prescription rate for cyclosporine (7,244 cases), its average per-patient expense and average expense per case was the highest among all medications at $296.90 and $67.34, respectively (Table 5).

The 9-year trend in medication prescriptions for AD treatment showed a decrease in the use of topical agents, with a particularly sharp decrease in 2015–2017, while there was a steady trend in use of adjuvant medications, optional medications, and other items (Fig 4). The trends for inpatient prescription and number of claims for prescribed medications are shown in S4 Table.

## Discussion

The incidence of AD in Korea in increasing, but relevant data on which policies can be based are rare. In this study, the incidence of AD plateaued throughout the 9-year study period from 2010–2018. These findings are consistent with a previous study reporting that the average annual prevalence of AD has plateaued at 2.6% over a 6-year period (2009–2014) [40]. The number of claims also remained at a similar level, but the total expense has increased each year, independent of WM or KM. With respect to sex distribution, there were more females who visited a medical institution for AD (142,334 (52.71%) vs 127,674 (47.29%)), with the number being approximately 1.1 times higher. This is consistent with a previous finding [40] that AD was more prevalent in females (20.10) than in males (17.83) in 2014 [12]. Meanwhile, prevalence was highest in the <5 years group (31.4%), followed by the 5–14 years (23.53%)

**Table 5. Outpatient medication prescriptions per classification.**

| Category | | | No. of claims | | Average cost per claim | Average cost per patient |
|---|---|---|---|---|---|---|
| | | | n | % | | |
| Basic prescription medications—topical agents | Topical corticosteroids | | 340,187 | 53.81% | 3.95 | 7.09 |
| | Topical calcineurin inhibitors | | 33,525 | 5.30% | 18.47 | 31.21 |
| Adjuvant prescription medications | Antihistamine agents | | 534,597 | 84.55% | 1.96 | 6.59 |
| | Skin infection control | Antibiotics for topical use & for systemic use | 37,668 | 5.96% | 3.27 | 5.54 |
| | | Antifungals for topical use & antimycotics for systemic use | 5,879 | 0.93% | 11.56 | 17.74 |
| | Gamma-linolenic acid | | 43,332 | 6.85% | 8.71 | 21.66 |
| Optional prescription medications | Immunosuppressants | Oral steroids | 163,811 | 25.91% | 0.71 | 1.46 |
| | | Cyclosporine | 7,244 | 1.15% | 67.34 | 296.90 |
| | Leukotriene antagonists: Phosphodiesterase inhibitor | | 9,195 | 1.45% | 9.43 | 18.51 |
| | Other therapeutics | Peptic ulcer solvent | 135,367 | 21.41% | 1.30 | 2.84 |
| | | Antitussive expectorants | 23,247 | 3.68% | 1.06 | 1.88 |
| | | Unclassified | 8,237 | 1.30% | 1.00 | 1.26 |

All expenses are converted from KRW to USD according to the annual average exchange rate (see S1 Table).

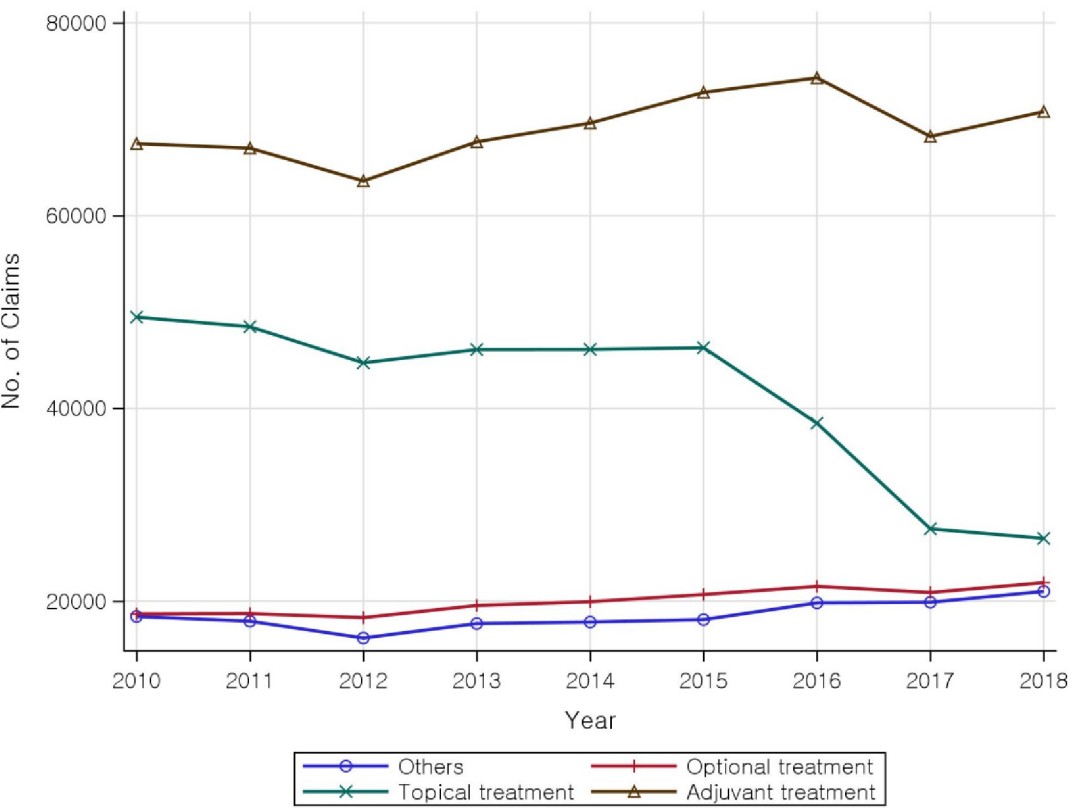

**Fig 4. Nine-year trends in prescribed medications for the treatment of atopic dermatitis.**

and 15–24 years groups (15.33%), indicating that AD is more common in infants, children, and adolescents. These findings are consistent with those of a 2014 study in which AD was more prevalent in children, with a prevalence rate of 95.0% in those aged <10 years [40]. This indicates that children have a predilection for AD [12]. Further, the number of patients aged 15–24 years was almost two times higher than that of patients aged 25–34 years (41,384 vs 23,091). Patients aged ≥55 years accounted for <10% of all patients, further supporting the above results. With respect to trends by age, there was a gradual decrease in the number of patients aged <14 years, while the number of patients aged ≥25 years increased (Fig 3). This is consistent with a previous study's finding of a sharp decrease in the number of AD patients aged <5 years in 2008–2017 and an increase in patients aged 6–18 years and ≥60 years [41].

With respect to healthcare usage, the number of patients who used WM was much higher than the number of patients who used KM. Among the WM patients, most were from primary medical institutions (clinics, 86.48%; tertiary/general/regular hospitals, 13.51%). The majority of healthcare users were in the age of <5 years (31.40%), 5–14 years (23.53%), and 15–24 years (15.33%), which may be attributed to the overall high prevalence of AD in the above age groups. There was no significant difference in healthcare usage by sex. The utilization trends by age groups and sex were similar across the types of medicine (WM, KM and WM+KM). The types of medical institutions, however, varied across the types of medicine; in specific, most KM utilization has taken place mostly in primary care, while more than 13% of WM utilization has taken place in tertiary care. For inpatient and outpatient care, the majority were outpatient cases (99.9% vs 0.06%).

Consistent with the results of previous studies, allergic rhinitis was a common comorbidity. There were no comorbidities other than allergic rhinitis, unspecified (J304), and functional dyspepsia (K30). Consistent with an earlier study's results [12], several WM and KM patients had skin-related diagnoses (L codes) and musculoskeletal disorders (M codes). For average annual expense by item, the highest expenses were for consultation, testing, treatment/surgery, other treatment, and injection fees in WM. In KM, they were injection which includes acupuncture and moxibustion, consultation, and hospitalization fees. Overall, the per-patient expense was 3–4 times higher in KM. For type of treatment, subcutaneous, intramuscular or intravenous injection accounted for 88,764 claims in WM. The most commonly used medication for injection therapy was hydroxine, followed by chlorphenamine, dexamethasone, betamethasone, diphenylpyraline, and triamcinolone, except for 2016. Among these, antihistamines (hydroxine, chlorphenamine, and diphenylpyraline) accounted for the highest number of inpatient prescriptions at 23.54%, followed by steroids (dexamethasone, betamethasone, and triamcinolone) at 9.16%.

Testing fees, which had the second highest number of claims in WM, mostly included allergy (specific IgE test), blood, and skin tests. Some patients did not present with elevated serum total or allergen-specific IgE levels, which is a common symptom of AD that helps to differentiate between non-IgE-related (intrinsic) AD and IgE-related (extrinsic) AD [23]. Allergy tests were performed in 12,824 cases over 9 years and incurred the highest average expense per case and average annual per-patient expense. In an earlier study, 87.0% of patients recognized the need for an allergy test, but the test was only administered in 59.9% of patients. The low rate was attributed to lack of manpower and facilities, irrelevance to actual treatment guidelines, preference for medication treatment, and difficulty in explaining the test results [23]. Our findings indicate that cost is an important factor influencing allergy testing rates.

For non-pharmaceutical therapies, the most used was photodynamic therapy, followed by wet-wrap therapy, phototherapy, and nebulizer treatment. Photodynamic therapy is used to regulate the severity and symptoms of patients with red blood cell AD [42]. Wet-wrap therapy is a method for quickly decreasing AD severity and is also used in severe rashes or intractable

AD [19]. Phototherapy, including natural light, NB UVB, BB-UVB, UVA, psoralen and long-wave UVA, UVA and UVB, and Goeckerman8, have also been reported to be beneficial for controlling AD [19, 42]. Phototherapy is used as second-line treatment in patients who fail first-line treatment with emollients, TCS, and TCI. It can also be used as maintenance therapy for patients with chronic disease [42]. Previous studies have reported that UV therapy can be effective for patients with severe symptoms who do not respond well to topical moisturizer treatment [19, 29].

Injection fees incurred the highest cost among KM items, which could be because injection was the primary modality in KM. This item includes acupuncture, cupping, and moxibustion. Acupuncture was performed in 71,185 cases. Cupping, including dry and bloodletting cupping, was performed in 12,827 cases. Acupuncture was reported to improve AD symptoms, markedly reducing type 1 hypersensitivity itch [29, 43–46]; one previous study reported that cupping therapy is effective in reducing itchiness and lichenification in AD [47], although further study is warranted. Several herbal concoctions have been presented as potential options for AD [48–50], although further clinical studies are required to confirm the effectiveness. In this study, CAM therapies such as herbal decoctions were not included in the database since HIRA-NPS data only include medical services covered by national health insurance.

With respect to medication treatment, the highest usage was for adjuvant treatment, followed by topical and optional medication treatment. The most commonly used adjuvant, topical, and optional medications were antihistamines (86.02%), TCS (91.03%), and oral steroids (47.19%), respectively. Notably, the use of TCS decreased, while the use of oral steroids increased. Additional studies are needed to determine the cause of the decrease in the use of TCS, which is the primary medication for AD treatment. The increased usage of oral steroids could be due to the increasing incidence of moderate-to-severe AD in Korea [2, 51–56].

Among primary medications, TCS was used in 45,871 cases in 2010, but its use decreased by more than half to 22,604 cases in 2018. Meanwhile, TCI was used in 3,602 cases in 2010, and its use increased by approximately 8.5% to 3,910 cases in 2018. The decreased usage of TCS could be the result of preventive measures against skin AEs like skin atrophy and telangiectasis, and in extremely rare cases systemic responses in the immune system [57, 58]. Further, compared with no treatment, topical and systemic steroid treatment is associated with a three times higher risk of psychological problems [59]. Among optional medications, oral steroids were used in 17,701 cases in 2010, and the usage increased steadily by approximately 8% to 19,251 cases in 2018. Oral steroids should only be used in the short term during acute AD exacerbation or in patients resistant to other treatment [58] because although it can markedly improve AD, it can also cause severe rebound flaring on discontinuation [58]. Cyclosporine was used in 484 cases in 2010, but its use increased sharply, by 2.7 times, to 1,312 cases in 2018. Cyclosporine is effective for treating AD, with most patients showing drastic reductions in disease activity within 2–6 weeks of commencement of treatment [60]. A previous study found a reduction in size of surface involvement and inflammation level in remaining dermatitis 6 weeks after the initiation of cyclosporine treatment [61]. Cyclosporine is the only systemic immunosuppressant approved by the US Food and Drug Administration covered by insurance in Korea [62]. However, although cyclosporine is the first–line medication of choice for severe AD not controlled by topical agents alone, cyclosporine is not actively prescribed [21].

In adjuvant medication usage, antihistamines showed an increasing trend, except in 2010–2012 and 2016–2017. H1 antihistamines with sedative action, such as hydroxyzine and diphenhydramine, are effective for AD, effectively controlling severe itchiness causing sleep disturbance. Meanwhile, non-sedative antihistamines have almost no effect on itch control in AD.

Recently developed second-generation antihistamines are expected to be effective owing to their anti-allergic action [58]. Dupilumab, an IL4Rα antagonist, has garnered attention as an innovative novel medication for AD. Importantly, it is an FDA-approved biologic for the treatment of pediatric AD [63]. However, in Korea, dupilumab has only been covered by insurance since January 1, 2020, and was rarely used prior to coverage due to high cost. Accordingly, its use was not identified in the analyzed data.

## Limitations and strengths

First, although the present study analyzed the healthcare usage data for AD in Korea, it was difficult to conclude that such healthcare usage was strictly for AD based on the disease classification system currently used by the HIRA. Thus, we included only patients with AD as the primary diagnosis; however, treatments for AD and for other diseases could not be completely differentiated. Second, we used annual data, not continuous data. Consequently, patient follow-up was limited to the same year. Third, treatment cost is commonly divided into three categories (direct, indirect, and intangible costs), but this study only analyzed treatment costs directly incurred from medical services received, excluding non-medical costs. KM claims are significantly fewer than WM claims, but there are several non-covered items [64]. Therefore, it is difficult to accurately account for all KM treatment for patients with AD with only KM claims data.

Despite these limitations, the present study also had the following strengths. First, to our knowledge, no studies have compared the healthcare usage among patients by year at the national level, and this is the first study to use the HIRA-NPS data (2010–2018) to analyze the healthcare usage among patients with AD. Second, the present study accounted for the unique nature of the dual healthcare system in Korea by comparing WM and KM. Third, this study was the first to divide medication treatment for AD into basic, adjuvant, and optional treatments and compare the frequencies. Lastly, this study analyzed the method of visit in terms of age and sex to provide data on the medical institutions visited by different sex and age groups.

## Conclusions

The incidence of AD in Korea has plateaued, but the total expense has increased, although by a small margin. The most common WM and KM treatments were injections and acupuncture, respectively. TCS was the most commonly used medication treatment, but its use is decreasing. The data on treatment trends and the associated costs provide important evidence to guide the insurance coverage system, and, importantly, to treat and manage AD. Moreover, the findings provide baseline evidence for developing national health policies for AD and related diseases.

## Supporting information

**S1 Table. Annual average KRW-USD exchange rate and price level of health expense.**
(DOCX)

**S2 Table. Basic characteristics of Korean and Western medicine usage.**
(DOCX)

**S3 Table. Top 10 WM/KM sub-diagnoses.**
(DOCX)

**S4 Table. Inpatient prescription.**
(DOCX)

## Author Contributions

**Conceptualization:** Sowon Kim, Ye-Seul Lee, Jiyoon Yeo, In-Hyuk Ha.

**Data curation:** Ye-Seul Lee, Jiyoon Yeo.

**Formal analysis:** Ye-Seul Lee, Jiyoon Yeo.

**Funding acquisition:** In-Hyuk Ha.

**Investigation:** Sowon Kim, Donghyo Lee.

**Methodology:** Donghyo Lee, In-Hyuk Ha.

**Project administration:** Donghyo Lee, In-Hyuk Ha.

**Resources:** Ye-Seul Lee, Jiyoon Yeo, Donghyo Lee.

**Supervision:** Donghyo Lee, In-Hyuk Ha.

**Validation:** Dong Kun Ko, In-Hyuk Ha.

**Writing – original draft:** Sowon Kim, Ye-Seul Lee, Jiyoon Yeo.

**Writing – review & editing:** Sowon Kim, Ye-Seul Lee, Donghyo Lee, Dong Kun Ko, In-Hyuk Ha.

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
