## [Decision Letter · Decision Letter 0]

12 Dec 2022

PONE-D-21-39525Healthcare utilization for atopic dermatitis: an analysis of the 2010–2018 Health Insurance Review and Assessment Service National Patient Sample DataPLOS ONE

Dear Dr. Ha,

Thank you for submitting your manuscript to PLOS ONE. After careful consideration, we feel that it has merit but does not fully meet PLOS ONE’s publication criteria as it currently stands. Therefore, we invite you to submit a revised version of the manuscript that addresses the points raised during the review process.

We look forward to receiving your revised manuscript.

Kind regards,

Dong Keon Yon, MD, FACAAI

Academic Editor

PLOS ONE

Journal Requirements:

"This research was funded by Jaseng Medical Foundation, Republic of Korea, grant number JS-RP-2021-21."

**Additional Editor Comments:**

Thank you for submitting your manuscript. The reviewers and I believe it is of potential value for our readers. However, the reviewers have raised a number of very important issues, and their excellent comments will need to be adequately addressed in a revision before the acceptability of your manuscript for publication in the Journal can be determined. We cannot guarantee that your revised paper will be chosen for publication; this would be solely based on how satisfactorily you have addressed the reviewer comments.

Reviewers' comments:

Reviewer's Responses to Questions

**Comments to the Author**

1. Is the manuscript technically sound, and do the data support the conclusions?

Reviewer #1: Yes

Reviewer #2: Yes

Reviewer #3: Yes

2. Has the statistical analysis been performed appropriately and rigorously? 

Reviewer #1: Yes

Reviewer #2: Yes

Reviewer #3: Yes

3. Have the authors made all data underlying the findings in their manuscript fully available?

Reviewer #1: Yes

Reviewer #2: Yes

Reviewer #3: Yes

4. Is the manuscript presented in an intelligible fashion and written in standard English?

Reviewer #1: Yes

Reviewer #2: Yes

Reviewer #3: Yes

5. Review Comments to the Author

Reviewer #1: Overall it is interesting to compare the use of Western vs Korean Medicine in such a large cohort. Be careful with making broad claims about effectiveness or adverse effects in the introduction and discussion. The potential harms of CAM needs to be addressed. Could be considered for review if some of these issues are addressed and suggest specific changes as below:

Some of the language used could be improved and rewritten. Some examples:

In the abstract:

Line 28-29:

This sentence is vague and could be improved by including specific treatments. Which intramuscular, subcut and IV injections?:

The most used Western medicine treatments included intramuscular, subcutaneous, and intravenous injections, while it was acupuncture therapy in Korean medicine.

Line 30:

Perhaps use the word prescription medication rather than drug:

The most used drug was topical corticosteroid, but its frequency of use also showed a decreasing trend.

Line 30-32:

Could you expand on the relevance of the study ie. How many countries use acupuncture to treat atopic dermatitis so this study may only be of particular relevance to those countries.

The healthcare usage and trends in Western medicine and Korean medicine treatment for atopic dermatitis in this study

provide baseline data for health policy makers and clinicians.

Line 35-36:

“Atopic dermatitis (AD) commonly occurs in childhood and is characterized by chronically relapsing rashes and itching”

Could be reworded to something like:

Atopic Dermatitis (AD) is an inflammatory chronic and relapsing condition that presents with erythematous, scaly, pruritic rashes. It is most common in childhood, however all age groups can be affected.

Line 36-37

I’m not sure that recent evidence has shown that it is occurs in older age groups, I believe older age groups have been known for some time.

Line 53:

I’m not sure what you mean by AD can cause localized eczema?

Line 60:

“However, an optimal treatment strategy that is individualized according to disease severity, has not been established.” - ?in korea, there are multiple clinical practice guidelines in other countries

When discussing CAM – you need to address the risks and side effects of CAM such as side effects from herbal medicines including systemic toxicities, allergic contact dermatitis, cost, risk of avoiding Western or conventional medicine and hence risk of leaving disease untreated, worsening of disease and risk of complications such as infection or scarring.

The different insurance types need to be explained for the non-Korean audience.

Line 219: CBC ?complete blood count, is this written in full earlier, otherwise be careful of abbreviations.

Line 232-233:

“This was followed by Gamma-linolenic acid, Staphylococcus aureus, and Pityrosporum ovale.”

I don’t think this sentence makes sense. It indicates staph aureus is a treatment when it is a bacterium?

Line 235-236:

What does optimal drug treatments mean?

Line 310-311

Herbal concoctions, such as Oryeong-san [39], GagamBangpungtongseong-san [40], and Gammakdaejo311tang [41] are effective in treating AD.

This is a broad sweeping claim. I don’t think that one case study and two animal studies can demonstrate effectiveness.

Line 320-321:

Systemic toxicities such as cushings syndrome are rare, and linked to inappropriate and overuse and this needs to be discussed.

I don’t believe there is a risk of lymphoma and TCS use, https://pubmed.ncbi.nlm.nih.gov/25840730/

Line 322:

Reframe this sentence to reference what this study showed. Saying that TCS increase psychological problems is a broad sweeping statement and may not apply to all cohorts. Otherwise this needs to be demonstrated with more evidence and references:

“compared with no treatment, topical and systemic steroid treatment is associated with a three times higher risk of psychological problems [50].”

340-341:

Might be worth mentioning upadacitinib here as a treatment option too.

Reviewer #2: I have no further comments regarding this, this is a standard, good paper, not overly exciting, however it will possibly find it's purpose in everyday practice. Although it is to an extent common wisdom, it is well organized, has a point and shows in a systematic manner the idea how to improve access to health care utilization in AD patients

Reviewer #3: Dear authors,

I have now completed the review of the manuscript titled "Healthcare utilization for atopic dermatitis: an analysis of the 2010–2018 Health Insurance Review and Assessment Service National Patient Sample Data."

In the present study, the authors analyzed the distribution and healthcare usage patterns of patients with atopic dermatitis using the 2010-2018 Health Insurance Review and Assessment Service data.

The manuscript is interesting and, in general, fair written.

I would like to recommend an "accept" decision, but before doing so, I would like to suggset one minor suggestion.

In the ‘Statistical analysis’ section, there are no citing references, therefore I would like to suggest adding the following reference in the sentence:

The average annual log change for each item, and the differences between WM and KM, were investigated[1, 2].

[1] https://doi.org/10.54724/lc.2022.e1

[2] https://doi.org/10.54724/lc.2022.e3

I hope good luck to the authors' future work and research.

Thank you.

6. PLOS authors have the option to publish the peer review history of their article (what does this mean?). If published, this will include your full peer review and any attached files.

Reviewer #1: No

Reviewer #2: No

Reviewer #3: No

---

## [Author Response · Author response to Decision Letter 0]

27 Feb 2023

Additional Editor Comments:

Thank you for submitting your manuscript. The reviewers and I believe it is of potential value for our readers. However, the reviewers have raised a number of very important issues, and their excellent comments will need to be adequately addressed in a revision before the acceptability of your manuscript for publication in the Journal can be determined. We cannot guarantee that your revised paper will be chosen for publication; this would be solely based on how satisfactorily you have addressed the reviewer comments.

- We would like to thank the Editor for giving us a chance to revise the manuscript. We believe we’ve addressed all of the reviewer’s comments and, thanks to the thoughtful comments, the overall quality of the manuscript has improved. 

Reviewers' comments:

Reviewer #1: Overall it is interesting to compare the use of Western vs Korean Medicine in such a large cohort. Be careful with making broad claims about effectiveness or adverse effects in the introduction and discussion. The potential harms of CAM needs to be addressed. Could be considered for review if some of these issues are addressed and suggest specific changes as below:

Some of the language used could be improved and rewritten. Some examples:

In the abstract:

Line 28-29:

This sentence is vague and could be improved by including specific treatments. Which intramuscular, subcut and IV injections?:

The most used Western medicine treatments included intramuscular, subcutaneous, and intravenous injections, while it was acupuncture therapy in Korean medicine.

- We appreciate the reviewer’s thoughtful comment. We revised the manuscript as follows:

- The most used Western medicine treatments were injections and oral medications involving topical corticosteroids, antihistamine agents, and oral steroids, while it was acupuncture therapy in Korean medicine. The frequency of the most frequently prescribed medication, topical corticosteroid, showed a decreasing trend over time.

Line 30:

Perhaps use the word prescription medication rather than drug:

The most used drug was topical corticosteroid, but its frequency of use also showed a decreasing trend.

- We appreciate the reviewer’s thoughtful comment. We revised the manuscript as follows:

- The frequency of the most frequently prescribed medication, topical corticosteroid, showed a decreasing trend over time.

Line 30-32:

Could you expand on the relevance of the study ie. How many countries use acupuncture to treat atopic dermatitis so this study may only be of particular relevance to those countries.

The healthcare usage and trends in Western medicine and Korean medicine treatment for atopic dermatitis in this study provide baseline data for health policy makers and clinicians.

- We appreciate the reviewer’s thoughtful comment. We revised the manuscript in the Abstract and in Introduction as follows:

Abstract

The findings in this study will inform healthcare policy makers and clinicians across different countries on the usage trends of Western medicine and Korean medicine treatment.

Introduction

Furthermore, a recent review found that the effectiveness of acupuncture in atopic dermatitis is being explored in other countries including China and Germany.

Line 35-36:

“Atopic dermatitis (AD) commonly occurs in childhood and is characterized by chronically relapsing rashes and itching”

Could be reworded to something like:

Atopic Dermatitis (AD) is an inflammatory chronic and relapsing condition that presents with erythematous, scaly, pruritic rashes. It is most common in childhood, however all age groups can be affected.

- We appreciate the reviewer’s thoughtful comment. We revised the manuscript as follows:

- Atopic dermatitis (AD) is a chronic and relapsing inflammatory condition that presents with erythematous, scaly, itchy rashes. It is most prevalent in children, but all age groups can be affected [1, 2]. 

Line 36-37

I’m not sure that recent evidence has shown that it is occurs in older age groups, I believe older age groups have been known for some time.

- We appreciate the reviewer’s thoughtful comment. We revised the manuscript as follows:

- Recent evidence indicates that it causes changes in the skin barrier and impairs immune function, increasing vulnerability to this disease throughout lifetime [3, 4].

Line 53:

I’m not sure what you mean by AD can cause localized eczema?

- We appreciate the reviewer’s thoughtful comment. We revised the manuscript as follows:

- From a personal health perspective, it can also have a negative impact by causing chronic skin diseases such as urticaria and alopecia; sleep disturbance due to itching; as well as complications of comorbid asthma and food allergies [18].

Line 60:

“However, an optimal treatment strategy that is individualized according to disease severity, has not been established.” - ?in korea, there are multiple clinical practice guidelines in other countries

- We appreciate the reviewer’s thoughtful comment. We revised the manuscript as follows:

- However, a long-term therapy plan that avoids corticosteroid adverse effects while enhancing their efficacy has always been the subject of controversy. Although more aggressive therapies involving systemic immunosuppressive medications, e.g., cyclosporine and mycophenolate, phototherapy, and allergen-immunotherapy is recommended for severe AD, an actual application and compliance in clinical settings have been reported to be relatively low.

When discussing CAM – you need to address the risks and side effects of CAM such as side effects from herbal medicines including systemic toxicities, allergic contact dermatitis, cost, risk of avoiding Western or conventional medicine and hence risk of leaving disease untreated, worsening of disease and risk of complications such as infection or scarring.

- We appreciate the reviewer’s thoughtful comment. We revised the manuscript as follows:

- Furthermore, potential risk of inappropriate use of CAM which may induce allergic reactions, additional burden of cost, and possible negligence of conventional medication by patients which may accelerate disease progress needs to be taken into account when analyzing utilization of CAM.

The different insurance types need to be explained for the non-Korean audience.

- We appreciate the reviewer’s thoughtful comment. We revised the manuscript as follows:

- The single insurer system in Korea by National Health Insurance (NHI), which supports 98% of all population and also provides data for 2% of the population who receive Medical Aid, provides healthcare coverage in both Western and Korean Medicine due to its unique dual healthcare system. Therefore, examining the NHI database provides knowledge on healthcare spendings from societal perspective in both Western and Korean Medicine.

Line 219: CBC ?complete blood count, is this written in full earlier, otherwise be careful of abbreviations.

- We appreciate the reviewer’s thoughtful comment. We revised the manuscript as follows:

- The second and third highest average per-patient expense was for other tests ($83.42) and complete blood count (CBC) test (Code B1~: $29.22), respectively, indicating that testing fees accounted for a large portion of the cost burden for patients.

Line 232-233:

“This was followed by Gamma-linolenic acid, Staphylococcus aureus, and Pityrosporum ovale.”

I don’t think this sentence makes sense. It indicates staph aureus is a treatment when it is a bacterium?

- We appreciate the reviewer’s thoughtful comment. We revised the manuscript as follows:

- . This was followed by antibiotics for topical use & for systemic use, antifungals for topical use & antimycotics for systemic use, and gamma-linolenic acid. The highest average expense per case was for antifungals for topical use & antimycotics for systemic use, while the highest average per-patient expense was for gamma-linolenic acid.

Line 235-236:

What does optimal drug treatments mean?

- We appreciate the reviewer’s thoughtful comment. We revised the manuscript and changed the term “drug treatments” to prescribed medications or prescription medication. Examples of the changes are as follows:

- Among optional prescription medications, oral steroids showed the highest number of claims at 163,811, accounting for approximately 47% of all optional prescriptions.

Line 310-311

Herbal concoctions, such as Oryeong-san [39], GagamBangpungtongseong-san [40], and Gammakdaejo311tang [41] are effective in treating AD.

This is a broad sweeping claim. I don’t think that one case study and two animal studies can demonstrate effectiveness.

- We appreciate the reviewer’s thoughtful comment. We revised the manuscript as follows:

- Herbal concoctions, such as Oryeong-san [41], GagamBangpungtongseong-san [42], and Gammakdaejo-tang [43] have been presented as potential options for AD, although further clinical studies are required to confirm the effectiveness.

Line 320-321:

Systemic toxicities such as cushings syndrome are rare, and linked to inappropriate and overuse and this needs to be discussed.

I don’t believe there is a risk of lymphoma and TCS use, https://pubmed.ncbi.nlm.nih.gov/25840730/

- We appreciate the reviewer’s thoughtful comment. We revised the manuscript as follows:

- The decreased usage of TCS could be a result of preventive measures against skin AEs like skin atrophy and telangiectasis; further, systemic AEs like hypothalamic-pituitary-adrenal inhibition and Cushing’s syndrome have also been suggested, although it’s extremely rare and often linked to overuse [50, 51].

Line 322:

Reframe this sentence to reference what this study showed. Saying that TCS increase psychological problems is a broad sweeping statement and may not apply to all cohorts. Otherwise this needs to be demonstrated with more evidence and references:

“compared with no treatment, topical and systemic steroid treatment is associated with a three times higher risk of psychological problems [50].”

- We appreciate the reviewer’s thoughtful comment. Upon second assessment of our manuscript, we decided that this sentence and reference does not adequately support the context in the paragraph and therefore removed them. 

340-341:

Might be worth mentioning upadacitinib here as a treatment option too.

- We appreciate the reviewer’s thoughtful comment. We revised the manuscript as follows:

- Recent findings indicate upadacitinib as an option for moderate to severe atopic dermatitis[56]; however, it has been covered by NHI since May of 2022 in Korea, therefore it was not included in this study.

 

Reviewer #2: I have no further comments regarding this, this is a standard, good paper, not overly exciting, however it will possibly find it's purpose in everyday practice. Although it is to an extent common wisdom, it is well organized, has a point and shows in a systematic manner the idea how to improve access to health care utilization in AD patients

- We thank the reviewer for the thoughtful comment.

Reviewer #3: Dear authors,

I have now completed the review of the manuscript titled "Healthcare utilization for atopic dermatitis: an analysis of the 2010–2018 Health Insurance Review and Assessment Service National Patient Sample Data."

In the present study, the authors analyzed the distribution and healthcare usage patterns of patients with atopic dermatitis using the 2010-2018 Health Insurance Review and Assessment Service data.

The manuscript is interesting and, in general, fair written.

I would like to recommend an "accept" decision, but before doing so, I would like to suggset one minor suggestion.

In the ‘Statistical analysis’ section, there are no citing references, therefore I would like to suggest adding the following reference in the sentence:

The average annual log change for each item, and the differences between WM and KM, were investigated[1, 2].

[1] https://doi.org/10.54724/lc.2022.e1

[2] https://doi.org/10.54724/lc.2022.e3

I hope good luck to the authors' future work and research. Thank you.

- We appreciate the reviewer’s thoughtful comment. We revised the manuscript as follows:

The average annual log change for each item, and the differences between WM and KM, were investigated[30, 31].

---

## [Decision Letter · Decision Letter 1]

12 Mar 2023

PONE-D-21-39525R1Healthcare utilization for atopic dermatitis: an analysis of the 2010–2018 Health Insurance Review and Assessment Service National Patient Sample DataPLOS ONE

Dear Dr. Ha,

Thank you for submitting your manuscript to PLOS ONE. After careful consideration, we feel that it has merit but does not fully meet PLOS ONE’s publication criteria as it currently stands. Therefore, we invite you to submit a revised version of the manuscript that addresses the points raised during the review process.

We look forward to receiving your revised manuscript.

Kind regards,

Dong Keon Yon, MD, FACAAI

Academic Editor

PLOS ONE

Additional Editor Comments:

Thank you for submitting your manuscript. The reviewers and I believe it is of potential value for our readers. However, the reviewers have raised a number of severe critical flaws, and their excellent comments will need to be adequately addressed in a revision before the acceptability of your manuscript for publication in the Journal can be determined.

Please remind that we cannot guarantee that your revised paper will be chosen for publication; this would be solely based on how satisfactorily you have addressed the reviewer comments.

Reviewers' comments:

Reviewer's Responses to Questions

**Comments to the Author**

1. If the authors have adequately addressed your comments raised in a previous round of review and you feel that this manuscript is now acceptable for publication, you may indicate that here to bypass the “Comments to the Author” section, enter your conflict of interest statement in the “Confidential to Editor” section, and submit your "Accept" recommendation.

Reviewer #1: (No Response)

Reviewer #3: All comments have been addressed

2. Is the manuscript technically sound, and do the data support the conclusions?

Reviewer #1: No

Reviewer #3: Yes

3. Has the statistical analysis been performed appropriately and rigorously? 

Reviewer #1: Yes

Reviewer #3: Yes

4. Have the authors made all data underlying the findings in their manuscript fully available?

Reviewer #1: No

Reviewer #3: Yes

5. Is the manuscript presented in an intelligible fashion and written in standard English?

Reviewer #1: Yes

Reviewer #3: Yes

6. Review Comments to the Author

Reviewer #1: While the authors revised some of their sections, they didn’t adequately address concerns that I had during my first review. Issues that need to be addressed: biased and generalized interpretation of results in discussion, not adequately contextualized the relevance of their study in the broader dermatology community. The study may have value in it’s objective results, however the write up fails to acknowledge existing literature and evidence in regards to CAM and other systemic treatments. One sentence on the damages of CAM (incorrectly worded as “inappropriate use of CAM”), does not adequately address this issue. There is significant and misinformed discussion about the side effects of conventional medicine which makes this paper appear biased and is promoting misinformation.

Line 60-63, this sentence is generalized and factually incorrect. This type of language, bias and poor representation of other treatment options is concerning. If there are studies that have found this, only one survey based study was referenced.

In addition, studies have found that although aggressive drug therapy involving systemic immunosuppressive drugs like cyclosporine and mycophenolate; phototherapy; and allergen-immunotherapy is recommended for severe AD, it is not actively complied with in actual clinical practice [20].

Some of the references cited as strong evidence eg reference 20, 24, 25 are survey based or low evidence studies. This is not adequately acknowledged in the write up.

Reference 48:

The conclusion of the study of reference 48 states “However, in the adjusted analysis, severity of AD was the main factor associated with an increased risk of lymphoma.”

The authors use this reference to support their statement that topical corticosteroids cause lymphoma. This is concerned that the evidence is misconstrued and promotes hysteria and misinformation about topical corticosteroids.

Reviewer #3: All comments have been addressed. Thank you to the authors and editors for considering my opinion on this manuscript.

7. PLOS authors have the option to publish the peer review history of their article (what does this mean?). If published, this will include your full peer review and any attached files.

Reviewer #1: No

Reviewer #3: No

---

## [Author Response · Author response to Decision Letter 1]

26 Apr 2023

Review Comments to the Author

Reviewer #1: While the authors revised some of their sections, they didn’t adequately address concerns that I had during my first review. 

Issues that need to be addressed: biased and generalized interpretation of results in discussion, not adequately contextualized the relevance of their study in the broader dermatology community. 

- We appreciate the reviewer’s comment. Based on the reviewer’s comment, we thoroughly revised the Introduction and paragraphs in the Discussion where CAM therapies were being discussed. Some of the examples are found below:

With respect to healthcare usage, the number of patients who used WM was much higher than the number of patients who used KM. Among the WM patients, most were from primary medical institutions (clinics, 86.48%; tertiary/general/regular hospitals, 13.51%). The majority of healthcare users were in the age of <5 years (31.40%), 5–14 years (23.53%), and 15–24 years (15.33%), which may be attributed to the overall high prevalence of AD in the above age groups. There was no significant difference in healthcare usage by sex. The utilization trends by age groups and sex were similar across the types of medicine (WM, KM and WM+KM). The types of medical institutions, however, varied across the types of medicine; in specific, most KM utilization has taken place mostly in primary care, while more than 13% of WM utilization has taken place in tertiary care. For inpatient and outpatient care, the majority were outpatient cases (99.9% vs 0.06%).

The study may have value in it’s objective results, however the write up fails to acknowledge existing literature and evidence in regards to CAM and other systemic treatments. 

- We appreciate the reviewer’s comment. Based on the reviewer’s comment, we thoroughly revised the sentences as follows:

Acupuncture was reported to improve AD symptoms, markedly reducing type 1 hypersensitivity itch [29, 43-46]; one previous study reported that cupping therapy is effective in reducing itchiness and lichenification in AD [47], although further study is warranted. Several herbal concoctions have been presented as potential options for AD [48-50], although further clinical studies are required to confirm the effectiveness. In this study, CAM therapies such as herbal decoctions were not included in the database since HIRA-NPS data only include medical services covered by national health insurance.

One sentence on the damages of CAM (incorrectly worded as “inappropriate use of CAM”), does not adequately address this issue. There is significant and misinformed discussion about the side effects of conventional medicine which makes this paper appear biased and is promoting misinformation.

- We appreciate the reviewer’s comment. Based on the reviewer’s comment, we thoroughly revised the sentences as follows and referenced the evidence with further accuracy:

While the effectiveness of CAM on AD has not been confirmed[26, 27], a few clinical trials and reviews have reported possible efficacy of acupuncture for symptom improvement and reduced recurrence across different populations [28-33], and one study reported a trend towards reduced TCS use in acupuncture and osteopathic treatment groups[34]. Nonetheless, it is crucial for both physicians and patients to note the uncertainties involved in CAM therapies due to lack of evidence on its safety and efficacy in AD management. The potential risk of inappropriate use of CAM which may induce allergic reactions, additional burden of cost, and possible negligence of conventional medication by patients which may accelerate disease progress needs to be taken into account when analyzing utilization of CAM.

Line 60-63, this sentence is generalized and factually incorrect. This type of language, bias and poor representation of other treatment options is concerning. If there are studies that have found this, only one survey based study was referenced.

- We appreciate the reviewer’s comment. Based on the reviewer’s comment, we thoroughly revised the sentences as follows:

Systemic immunosuppressants e.g., cyclosporine, methotrexate, and mycophenolate mofetil are recommended for severe AD, while caution is warranted to avoid side effects when long-term treatments are administered [19, 20]. One survey conducted to Korean dermatologists stated that systemic immunosuppressants are not actively used in clinical settings[21]. 

Some of the references cited as strong evidence eg reference 20, 24, 25 are survey based or low evidence studies. This is not adequately acknowledged in the write up.

- We appreciate the reviewer’s comment. Based on the reviewer’s comment, we thoroughly revised the sentences as follows:

While the effectiveness of CAM on AD has not been confirmed[26, 27], a few clinical trials and reviews have reported possible efficacy of acupuncture for symptom improvement and reduced recurrence across different populations [28-33], and one study reported a trend towards reduced TCS use in acupuncture and osteopathic treatment groups[34]. Nonetheless, it is crucial for both physicians and patients to note the uncertainties involved in CAM therapies due to lack of evidence on its safety and efficacy in AD management. The potential risk of inappropriate use of CAM which may induce allergic reactions, additional burden of cost, and possible negligence of conventional medication by patients which may accelerate disease progress needs to be taken into account when analyzing utilization of CAM.

Reference 48:

The conclusion of the study of reference 48 states “However, in the adjusted analysis, severity of AD was the main factor associated with an increased risk of lymphoma.”

The authors use this reference to support their statement that topical corticosteroids cause lymphoma. This is concerned that the evidence is misconstrued and promotes hysteria and misinformation about topical corticosteroids.

- We appreciate the reviewer’s comment. Based on the reviewer’s comment, we thoroughly revised the sentences and deleted the phrase that the reviewer pointed out. The paraphrased sentence is as follows:

The decreased usage of TCS could be the result of preventive measures against skin AEs like skin atrophy and telangiectasis, and in extremely rare cases systemic responses in the immune system [57, 58].

---

## [Decision Letter · Decision Letter 2]

17 May 2023

Healthcare utilization for atopic dermatitis: an analysis of the 2010–2018 Health Insurance Review and Assessment Service National Patient Sample Data

PONE-D-21-39525R2

Dear Dr. Ha,

We’re pleased to inform you that your manuscript has been judged scientifically suitable for publication and will be formally accepted for publication once it meets all outstanding technical requirements.

Kind regards,

Dong Keon Yon, MD, FACAAI

Academic Editor

PLOS ONE

Additional Editor Comments (optional):

This is an excellent paper.

Reviewers' comments:

Reviewer's Responses to Questions

**Comments to the Author**

1. If the authors have adequately addressed your comments raised in a previous round of review and you feel that this manuscript is now acceptable for publication, you may indicate that here to bypass the “Comments to the Author” section, enter your conflict of interest statement in the “Confidential to Editor” section, and submit your "Accept" recommendation.

Reviewer #1: All comments have been addressed

2. Is the manuscript technically sound, and do the data support the conclusions?

Reviewer #1: Yes

3. Has the statistical analysis been performed appropriately and rigorously? 

Reviewer #1: Yes

4. Have the authors made all data underlying the findings in their manuscript fully available?

Reviewer #1: Yes

5. Is the manuscript presented in an intelligible fashion and written in standard English?

Reviewer #1: Yes

6. Review Comments to the Author

Reviewer #1: Thank you for your revision. The authors have improved the bias in the article and addressed previous comments.

7. PLOS authors have the option to publish the peer review history of their article (what does this mean?). If published, this will include your full peer review and any attached files.

Reviewer #1: No

---

## [Editor Report · Acceptance letter]

16 Jun 2023

PONE-D-21-39525R2 

Healthcare utilization for atopic dermatitis: an analysis of the 2010–2018 Health Insurance Review and Assessment Service National Patient Sample Data 

Dear Dr. Ha:

I'm pleased to inform you that your manuscript has been deemed suitable for publication in PLOS ONE. Congratulations! Your manuscript is now with our production department. 

Kind regards, 

on behalf of

Dr. Dong Keon Yon 

Academic Editor

PLOS ONE